

# Cmah deficiency may lead to age-related hearing loss by influencing miRNA-PPAR mediated signaling pathway

Juhong Zhang[1], Na Wang[2] and Anting Xu[2,3]

[1] Department of Otolaryngology, Shanghai Jiao Tong University Affiliated Sixth People's Hospital South Campus, Southern Medical University Affiliated Fengxian Hospital, Shanghai, China
[2] Department of Otolaryngology/Head and Neck Surgery, the Second Hospital of Shandong University, Jinan, China
[3] NHC. Key Laboratory of Otorhinolaryngology, Shandong University, Jinan, China

Corresponding author
Anting Xu, antingxu@sdu.edu.cn

## ABSTRACT

**Background.** Previous evidence has indicated CMP-Neu5Ac hydroxylase (Cmah) disruption inducesaging-related hearing loss (AHL). However, its function mechanisms remain unclear. This study was to explore the mechanisms of AHL by using microarray analysis in the Cmah deficiency animal model.

**Methods.** Microarray dataset GSE70659 was available from the Gene Expression Omnibus database, including cochlear tissues from wild-type and Cmah-null C57BL/6J mice with old age (12 months, $n = 3$). Differentially expressed genes (DEGs) were identified using the Linear Models for Microarray data method and a protein–protein interaction (PPI) network was constructed using data from the Search Tool for the Retrieval of Interacting Genes database followed by module analysis. Kyoto Encyclopedia of Genes and Genomes pathway enrichment analysis was performed using the Database for Annotation, Visualization and Integrated Discovery. The upstream miRNAs and potential small-molecule drugs were predicted by miRwalk2.0 and Connectivity Map, respectively.

**Results.** A total of 799 DEGs (449 upregulated and 350 downregulated) were identified. Upregulated DEGs were involved in Cell adhesion molecules (ICAM1, intercellular adhesion molecule 1) and tumor necrosis factor (TNF) signaling pathway (FOS, FBJ osteosarcoma oncogene; ICAM1), while downregulated DEGs participated in PPAR signaling pathway (PPARG, peroxisome proliferator-activated receptor gamma). A PPI network was constructed, in which FOS, ICAM1 and PPARG were ranked as hub genes and PPARG was a transcription factor to regulate other target genes (ICAM1, FOS). Function analysis of two significant modules further demonstrated PPAR signaling pathway was especially important. Furthermore, mmu-miR-130b-3p, mmu-miR-27a-3p, mmu-miR-27b-3p and mmu-miR-721 were predicted to regulate PPARG. Topiramate were speculated to be a potential small-molecule drug to reverse DEGs in AHL.

**Conclusions.** PPAR mediated signaling pathway may be an important mechanism for AHL. Downregulation of the above miRNAs and use of topiramate may be potential treatment strategies for ALH by upregulating PPARG.

## INTRODUCTION

Hearing loss is the most common sensorineural deficit in the elderly, and it is estimated that 700 million persons have moderate to profound hearing loss worldwide in 2015, with approximately 30% of them occurred in their seventies and 50% in their eighties (*Niklaus, Dirk & Rudolf, 2011*; *Hjalte, Brännström & Gerdtham, 2012*; *Quaranta et al., 2015*). Age-related hearing loss (AHL) can lead to communication difficulties and cause social isolation, depression and anxiety, all of which severely influence the quality of life of patients (*Ciorba et al., 2012*). Furthermore, AHL is demonstrated to trigger cognitive function impairment in patients and thus may impose a large economic burden on families and society (*Peelle & Wingfield, 2016*). Thus, how to manage AHL has been an important public health issue.

Increasing evidence has indicated that oxidative stress is a crucial pathogenesis for AHL (*Fujimoto & Yamasoba, 2014*). Plasma reactive oxygen species (ROS) levels (i.e., hydrogen peroxide, hypochlorite and hydroxyl radicals) are observed to be significantly elevated (*Hwang et al., 2012*), while antioxidant retinol and zinc levels are significantly reduced in AHL patients (*Lasisi & Lasisi, 2015*). Linear regression reveals ROS and radical scavenger levels are positively and negatively associated with hearing thresholds of patients, respectively (*Hwang et al., 2012*; *Lasisi & Fehintola, 2011*). Furthermore, AHL animal model experiments also confirmed ROS excessively accumulated (*Riva et al., 2007*), but antioxidant enzymes [such as superoxide dismutase (SOD), reduced glutathione (GSH)/oxidized glutathione (GSSG)] strongly decreased in the cochlear spiral ganglion neurons and hair cells (*Coling et al., 2009*; *Menardo et al., 2012*). Even, AHL phenotype can be directly mimicked by selective knockout of SOD1 gene in mice (*Watanabe et al., 2014*). Oxidative stress may result in mitochondrial DNA mutations (*Markaryan, Nelson & Raul Hinojosa, 2009*; *Yamasoba et al., 2007*) and subsequently initiate BCl-2/Bax and caspase-3 mediated apoptotic pathways in the sensory cells and neurons of the cochlea (*Du et al., 2015*; *Huang et al., 2016*), which ultimately contribute to the development of hearing loss. Accordingly, supplementation of antioxidants (i.e., vitamin C, N-acetyl-cysteine) (*Ding et al., 2016*; *Kang et al., 2014*) or suppression of cell apoptosis of (i.e., caloric restriction, Erlong Zuoci decoction) (*Dong et al., 2016*; *Someya et al., 2010a*) may be underlying strategies to delay the onset of AHL and prevent pathological damages in the cochlea. However, the mechanisms of AHL remain not completely understood and current preventative or therapeutic interventions have not been universally acknowledged. Thereby, there is still a need to investigate the etiology of AHL to develop potential approaches for intervention of AHL.

CMP-Neu5Ac hydroxylase (Cmah) is an enzyme to catalyze the hydroxylation of N-acetylneuraminic acid (Neu5Ac) to N-glycoloylneuraminic acid (Neu5Gc). Neu5Gc is an important sialic acid and thus may play an important role for maintaining the structural and function of auditory system (*Go, Yoshikawa & Inokuchi, 2011*; *Inokuchi et al., 2017*). Cmah-deficient mice are shown to exhibit reduced hearing sensitivity in old age, accompanied with loss of sensory hair cells, spiral ganglion neurons, and/or stria vascularis degeneration throughout the cochlea (*Kwon et al., 2015*; *Hedlund et al., 2007*). These studies indicate mice with Cmah-null can act as a model for studying the mechanisms

of AHL in humans, which had been used in the study of *Kwon et al. (2015)*. Based on a high throughput microarray analysis technology, *Kwon et al. (2015)* demonstrated there were 631 up-regulated and 729 down-regulated genes in Cmah-null mice-derived cochlear tissues compared to control mice-derived cochlear tissues. Function enrichment analysis and PCR validation suggsted downregulated sirtuin deacetylase 3 (Sirt3), a mitochondrial NAD+-dependent deacetylase, may be involved in AHL via decreasing the expression of Fox1 and then promoting the production of ROS (*Kwon et al., 2015*). The key roles of Sirt3 for the development of AHL were also proved in the studies of other scholars (*Someya et al., 2010b*; *Zeng et al., 2014*). Therefore, manipulation of Sirt3 expression might represent a new approach to combat AHL. However, the use of Cmah-null mice to investigate the mechanism of AHL remains rarely reported.

The present study aimed to screen more crucial genes for explaining the mechanisms of AHL by re-analyzing the microarray data of *Kwon et al. (2015)* through addition of network-related bioinformatics algorithms. In addition, small molecule drugs were also predicted in order to find potential treatments for AHL.

## MATERIAL AND METHODS

### Microarray data

The microarray data under accession number GSE70659 were collected from the Gene Expression Omnibus (GEO) database (http://www.ncbi.nlm.nih.gov/geo/) (*Kwon et al., 2015*) (Supplemental Information 1), which contained cochlear tissues from wild-type (WT, $n = 3$) and Cmah-null ($n = 3$) C57BL/6J mice with old age (12 months).

### Data normalization and DEGs identification

The raw data (CEL files) downloaded from the Illumina MouseRef-8 v2.0 expression beadchip platform GPL6885 were preprocessed (including background adjustment, log2 transformation and quantile normalization) using the lumiR package in R (*R Core Team, 2017*). The DEGs between WT and Cmah-null mice were identified using the Linear Models for Microarray data (LIMMA) method (*Smyth, 2005*) in the Bioconductor R package (http://www.bioconductor.org/packages/release/bioc/html/limma.html). After the $t$-test, and the $p$-value was multiple corrected with the Benjamini–Hochberg (BH) procedure (*Benjamini & Hochberg, 1995*). Genes were considered to be significantly differentially at $p < 0.05$ and |logFC(fold change)| $> 0.5$ due to the poor BH-adjusted $p$-value.

### Protein–protein interaction (PPI) network construction

The PPI pairs were downloaded from acknowledged STRING 10.0 (Search Tool for the Retrieval of Interacting Genes; https://string-db.org/) database (*Szklarczyk et al., 2015*) and then the DEGs were imported into the PPI data to obtain the whole PPI network. The PPIs with combined scores $>0.4$ were selected to construct the PPI network which was visualized using the Cytoscape software (version 2.8; http://www.cytoscape.org/) (*Kohl, Wiese & Warscheid, 2011*). The crucial nodes within the PPI network were analyzed based on three topological properties using the CytoNCA plugin in Cytoscape software (http://apps.cytoscape.org/apps/cytonca) (*Tang et al., 2015*), including degree [the number

of interactions per node (protein)], betweenness (the number of shortest paths that pass through each node) and closeness centrality (the average length of the shortest paths to access all other proteins in the network). Functionally related and densely interconnected clusters were extracted from the large PPI network using the Molecular Complex Detection (MCODE) plugin of Cytoscape software according to the following parameters: degree cutoff = 5; node score cutoff = 0.4; k-core = 5; and maximum depth = 100 (ftp://ftp.mshri.on.ca/pub/BIND/Tools/MCODE) (*Bader & Hogue, 2003*). Modules were considered significant with MCODE score ≥ 4 and nodes ≥ 6.

Furthermore, whether the DEGs were transcription factors (TFs) and the TF-target gene interactions were predicted by the TRANSFAC database (http://www.gene-regulation.com/pub/databases.html) (*Matys et al., 2006*), and then were integrated into the PPI network to establish a regulatory network.

## Function enrichment analysis

Kyoto Encyclopedia of Genes and Genomes (KEGG) pathway enrichment analyses were performed to investigate the underlying functions of all DEGs and DEGs in the network using The Database for Annotation, Visualization and Integrated Discovery (DAVID) online tool (version 6.8; http://david.abcc.ncifcrf.gov) (*Huang, Sherman & Lempicki, 2009*). $P$-value <0.05 was set as the cut-off value.

## miRNA-target gene regulatory network construction

The miRNAs that can regulate the DEGs in the PPI network were predicted using the miRWalk database (version 2.0; http://www.zmf.umm.uni-heidelberg.de/apps/zmf/mirwalk2) with default significant parameters. Only the interaction relationships can be predicted by nine common algorithms, including miRWalk, Microt4, miRanda, miRDB, miRMap, miRNAMap, RNA22, RNAhybrid and Targetscan, were included to construct the miRNA-target gene regulatory network using the Cytoscape software (*Kohl, Wiese & Warscheid, 2011*).

## Screening of small-molecule drugs for treatment of AHL

The name of DEGs identified in the PPI network were converted to HG-U133A probe set IDs and uploaded to the Connectivity Map (CMAP, http://www.broadinstitute.org/cmap/) database which is a collection of genome-wide transcriptional expression data from human cancer cell lines treated with bioactive small molecules. If the enrichment score was close to −1, the corresponding small molecules were the potential drugs to reverse the expression of the query signature. Significant small-molecule drugs were selected according to the threshold value of $p < 0.05$ and |mean| >0.4.

## RESULTS

### Identification of DEGs

After data normalization, 799 genes were identified as DEGs between WT and Cmah-null mice based on the threshold of $p < 0.05$ and |logFC| >0.5, including 449 upregulated (such as Fos, FBJ osteosarcoma oncogene) and 350 downregulated genes (such as Ucp1, uncoupling

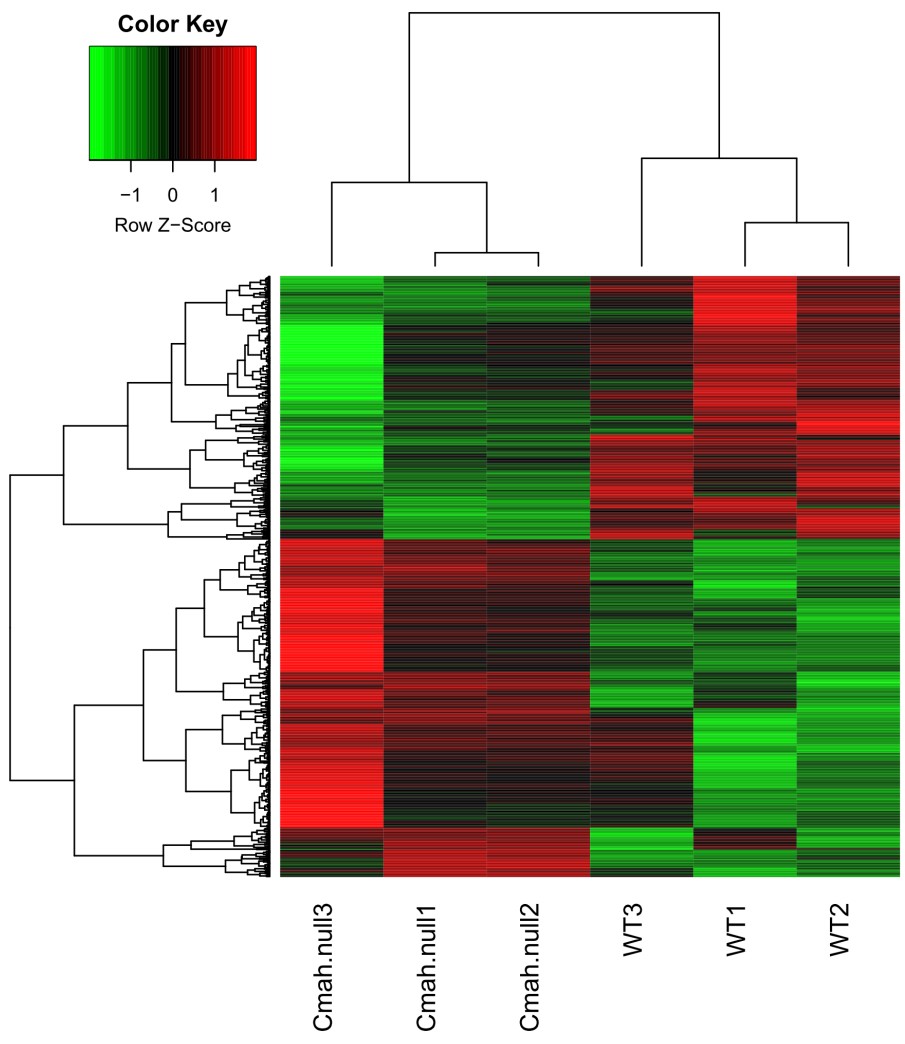

**Figure 1** Heat map of differentially expressed genes between Cmah-null and wild-type mice.

protein 1 (mitochondrial, proton carrier); Acadm, acyl-Coenzyme A dehydrogenase, medium chain). All the DEGs are listed in Supplemental Information 2. As shown in Fig. 1, heat map illustrated that the expression patterns of genes were obviously altered in Cmah-knockout mice compared with control.

## Function enrichment analysis of DEGs

The above differential genes were subjected to the DAVID for function enrichment analysis. As a result, 16 KEGG pathways were enriched for upregulated DEGs, including ECM-receptor interaction (LAMA1, laminin, alpha 1; ITGA5, integrin subunit alpha 5), Cell adhesion molecules (ICAM1, intercellular adhesion molecule 1; ITGA5), Focal adhesion (LAMA1; ITGA5), Cytokine-cytokine receptor interaction (TNFSF13B, TNF superfamily member 13b) and TNF signaling pathway (FOS; ICAM1); while 20 KEGG pathways were for downregulated DEGs, including Oxidative phosphorylation (UQCRC2,

ubiquinol cytochrome c reductase core protein 2), metabolism related and PPAR signaling pathway (ACADM; PPARG, Peroxisome proliferator-activated receptor gamma; UCP1; SCD1, stearoyl-Coenzyme A desaturase 1) (Table 1).

## PPI network construction

After mapping the DEGs into the protein interactions downloaded from the STRING database, a PPI network was constructed (Fig. 2) which included 587 nodes (303 upregulated and 284 downregulated; 52 TFs) and 2,944 edges (interaction relationships) (Supplemental Information 3). UQCRC2, SOD2 (superoxide dismutase 2), FOS, ICAM1 and PPARG were suggested to be hub genes by calculating the degree, betweenness, and closeness centrality of nodes in the PPI network. Among them, FOS and PPARG were TFs (Table 2) and they both interacted with ICAM1. Furthermore, PPARG also could interact with FOS and SOD2.

After cluster analysis according to the given parameters, two significant modules were obtained (Table 3). Function enrichment analysis showed that the genes in module 1 (Fig. 3) were closely related to Oxidative phosphorylation (UQCRC2), while the genes in module 2 (Fig. 4) were significantly enriched in metabolism related and PPAR signaling pathway (ACADM) (Table 4).

## miRNA–target gene regulatory network analysis

Using the miRWalk2.0 database, 82 DEGs were predicted to be regulated by 193 miRNAs in 9 databases (Supplemental Information 4). Then, 166 interaction relationships between 43 upregulated DEGs and 117 miRNAs as well as 182 interaction relationships between 39 downregulated DEGs and 76 miRNAs were used for constructing the upregulated (Fig. 5) and downregulated (Fig. 6) miRNA-mRNA regulatory networks, respectively. As shown in Fig. 5, FOS can be regulated by mmu-miR-221-3p or mmu-miR-222-3p, while PPARG can be regulated by mmu-miR-130b-3p, mmu-miR-27a-3p, mmu-miR-27b-3p and mmu-miR-721 in Fig. 6.

## Small-molecule drugs

The DEGs in the PPI network were uploaded into CMAP database to obtain the small-molecule drugs. As a result, 69 small-molecule chemicals with negative mean and enrichment scores were predicted, such as adiphenine, DL-PPMP, decitabine and topiramate. This finding indicated their potential ability to inhibit the development of AHL (Table 5).

## DISCUSSION

In the present study, Cmah-null mice were used as an animal model to investigate the underlying mechanisms of AHL. In line with the study of Kwon et al. (*Kwon et al., 2015*), oxidative phosphorylation pathway was enriched in the DEGs between Cmah-null mice and WT, and SOD2, SIRT3 were crucial genes in PPI network (Table 3), further demonstrating the oxidative stress pathogenesis of AHL (*Fujimoto & Yamasoba, 2014*). In addition, our current study also found ECM-receptor interaction (LAMA1), adhesion (ICAM1),
**Table 1  KEGG pathways for differentially expressed genes in the PPI network.**

|  | Term | *P*-value | Genes |
|---|---|---|---|
| UP | mmu04640:Hematopoietic cell lineage | 1.89E−05 | CD37, GP5, GP1BB, CD3E, ITGA5, CSF1, H2-EB1, ANPEP, ITGA3, ITGA2B… |
|  | mmu04512:ECM-receptor interaction | 2.85E−05 | VWF, LAMA1, CD47, LAMB3, GP5, GP1BB, ITGA5, ITGB4, ITGA3, ITGA2B… |
|  | mmu05164:Influenza A | 0.001774 | ICAM1, MYD88, HSPA2, SOCS3, IRF7, H2-EB1, PML, H2-AB1, OAS2, CCL5… |
|  | mmu05168:Herpes simplex infection | 0.002779 | FOS, MYD88, SOCS3, IRF7, H2-EB1, PML, PER2, PER1, H2-AB1, OAS2… |
|  | mmu05150:Staphylococcus aureus infection | 0.005378 | ICAM1, SELP, C4B, H2-EB1, H2-AB1, SELPLG |
|  | mmu04510:Focal adhesion | 0.007604 | VWF, LAMA1, LAMB3, ITGA5, RASGRF1, PAK4, ITGB4, ITGA3, ZYX, MYL12A… |
|  | mmu05217:Basal cell carcinoma | 0.008061 | BMP4, WNT10A, WNT7B, WNT4, WNT3A, WNT6 |
|  | mmu05200:Pathways in cancer | 0.008918 | BMP4, WNT10A, RALBP1, WNT3A, PML, FGF10, FOXO1, ITGA3, LAMA1, FOS… |
|  | mmu05323:Rheumatoid arthritis | 0.010803 | FOS, ICAM1, TNFSF13B, CSF1, H2-EB1, H2-AB1, CCL5 |
|  | mmu04060:Cytokine-cytokine receptor interaction | 0.024171 | OSM, CXCL14, TNFSF13B, PRLR, CSF1, CXCR1, CXCR2, CX3CL1, CCL5, BMP7… |
|  | mmu04151:PI3K-Akt signaling pathway | 0.029269 | EFNA1, CSF1, ITGB4, FGF10, ITGA3, CHAD, OSM, VWF, LAMA1, LAMB3… |
|  | mmu04611:Platelet activation | 0.02988 | VWF, ORAI1, GP5, GP1BB, PLCG2, MYL12A, ITPR3, ITGA2B |
|  | mmu04514:Cell adhesion molecules (CAMs) | 0.031608 | ICAM1, SIGLEC1, SELP, CLDN4, CLDN3, H2-EB1, H2-AB1, SELPLG, CLDN23 |
|  | mmu04668:TNF signaling pathway | 0.037976 | FOS, ICAM1, SOCS3, CSF1, CX3CL1, CCL5, JUNB |
|  | mmu04550:Signaling pathways regulating pluripotency of stem cells | 0.038071 | BMP4, WNT10A, WNT7B, WNT4, OTX1, WNT3A, WNT6, MEIS1 |
|  | mmu05205:Proteoglycans in cancer | 0.042265 | WNT10A, WNT7B, WNT4, TIAM1, ITGA5, WNT3A, PLCG2, ITPR3, WNT6, TWIST2 |
| Down | mmu00190:Oxidative phosphorylation | 6.68E−28 | UQCRC2, NDUFB3, ATP5E, NDUFB4, NDUFB5, NDUFB8, NDUFB9, COX7B, CYC1, NDUFB2… |
|  | mmu05012:Parkinson's disease | 1.48E−25 | UQCRC2, NDUFB3, ATP5E, NDUFB4, NDUFB5, NDUFB8, NDUFB9, COX7B, CYC1, NDUFB2… |
|  | mmu05016:Huntington's disease | 2.16E−25 | UQCRC2, NDUFB3, POLR2G, ATP5E, NDUFB4, NDUFB5, NDUFB8, NDUFB9, PPARG, COX7B… |
|  | mmu01100:Metabolic pathways | 1.11E−23 | UQCRC2, ATP5E, GNPDA2, CYC1, PDHB, CMBL, UQCR10, UQCR11, NDUFS4, IDH3G, MCEE… |
|  | mmu05010:Alzheimer's disease | 8.60E−22 | UQCRC2, NDUFB3, ATP5E, NDUFB4, NDUFB5, NDUFB8, NDUFB9, COX7B, CYC1, NDUFB2… |
|  | mmu04932:Non-alcoholic fatty liver disease (NAFLD) | 2.29E−21 | UQCRC2, NDUFB3, NDUFB4, NDUFB5, NDUFB8, NDUFB9, COX7B, CYC1, NDUFB2, UQCR10… |
|  | mmu01200:Carbon metabolism | 2.99E−11 | ALDH6A1, ACADM, ACO2, SUCLG1, ECHS1, FBP2, ACAT2, PDHB, SDHB, TPI1… |
|  | mmu01130:Biosynthesis of antibiotics | 2.59E−10 | ACAA2, ACADM, ACO2, SUCLG1, ECHS1, AK2, FBP2, ACAT2, PDHB, CMBL… |

**Table 1** (*continued*)

| Term | *P*-value | Genes |
|---|---|---|
| mmu00071:Fatty acid degradation | 1.21E−09 | ECI1, ECI2, ACAA2, ACADSB, CPT2, ACADM, ECHS1, ACADL, ACAT2, HADHB… |
| mmu00020:Citrate cycle (TCA cycle) | 2.22E−09 | SDHB, IDH3G, ACO2, SUCLG1, DLD, SDHD, IDH2, IDH1, PDHA1, FH1… |
| mmu00280:Valine, leucine and isoleucine degradation | 5.07E−09 | ACAA2, ALDH6A1, ACADSB, ACADM, ECHS1, ACAT2, HADHB, DBT, MCEE, DLD… |
| mmu01212:Fatty acid metabolism | 2.57E−08 | ACADVL, SCD1, ACAA2, ACADSB, ACSL1, ACADM, CPT2, ECHS1, ACADL, ACAT2… |
| mmu00640:Propanoate metabolism | 2.58E−06 | ALDH6A1, ACADM, SUCLG1, MCEE, ECHS1, ACAT2, PCCB, PCCA |
| mmu04260:Cardiac muscle contraction | 1.47E−05 | UQCRC2, UQCR10, CACNA2D1, UQCR11, COX8B, COX7A1, UQCRH, COX7B, CYC1, COX6B1… |
| mmu03320:PPAR signaling pathway | 2.07E−05 | SCD1, ACSL1, ACADM, CPT2, PPARG, FABP3, AQP7, UCP1, ACADL, ACSL5… |
| mmu04146:Peroxisome | 1.66E−04 | ECI2, ACSL1, ECH1, NUDT7, ABCD2, IDH2, IDH1, SCP2, SOD2, ACSL5 |
| mmu00630:Glyoxylate and dicarboxylate metabolism | 5.85E−04 | ACO2, MCEE, DLD, ACAT2, PCCB, PCCA |
| mmu01210:2-Oxocarboxylic acid metabolism | 0.010183 | IDH3G, ACO2, IDH2, IDH1 |
| mmu00620:Pyruvate metabolism | 0.014154 | DLD, PDHA1, FH1, ACAT2, PDHB |
| mmu00310:Lysine degradation | 0.036609 | EHMT1, HYKK, ECHS1, ACAT2, NSD1 |

**Notes.**

PPI, protein and protein interaction; KEGG, Kyoto Encyclopedia of Genes and Genomes.

inflammation (FOS, ICAM1, TNFSF13B) and PPAR signaling pathways (PPARG). Among them, PPAR signaling pathway may be especially important because the following causes: (1) this pathway was enriched for the genes in PPI and significant modules; (2) PPARG was a TF; and (3) PPARG could interact with SOD2, FOS and ICAM1. Furthermore, we predicted PPARG can be regulated by mmu-miR-130b-3p, mmu-miR-27a-3p, mmu-miR-27b-3p and mmu-miR-721. Also, adiphenine, DL-PPMP, decitabine and topiramate were speculated to be potential small-molecule drugs to reverse the expression of PPARG in AHL. Accordingly, we hypothesize downregulated PPARG may be involved in AHL by influencing adhesion, inflammation and oxidative stress, while downregulation of the above miRNAs and the use of the above small-molecule drugs may be potential treatment strategies for AHL by upregulating PPARG.

The extracellular matrix (ECM), the non-cellular component throughout all tissues and organs, and adherens junctions between cells and ECM are essential for maintenance of the structural and functional integrity of organs. ECM (i.e., laminin, integrin, fibronectin or collagen) and adherens (i.e., cadherin, syndecan-1, tenascin-C, Connexin or Icam) molecules are suggested to play a vital role for the growth and proliferation of cochlear sensorineural epithelial cells and sensory cell synaptogenesis (*Evans et al., 2007*; *Toyama et al., 2005*; *Wang, Hu & Yang, 2015*). The expression changes in the above molecules may cause hearing loss. For example, *Suzuki et al. (2005)* found type IX collagen knockout mice exhibited abnormal integrity of collagen fibers in the tectorial membrane and showed progressive hearing loss by auditory brainstem response assessment. *Gottesberge et al. (2008)* demonstrated deletion of the discoidin domain receptor 1 (DDR1) in mouse, a tyrosine

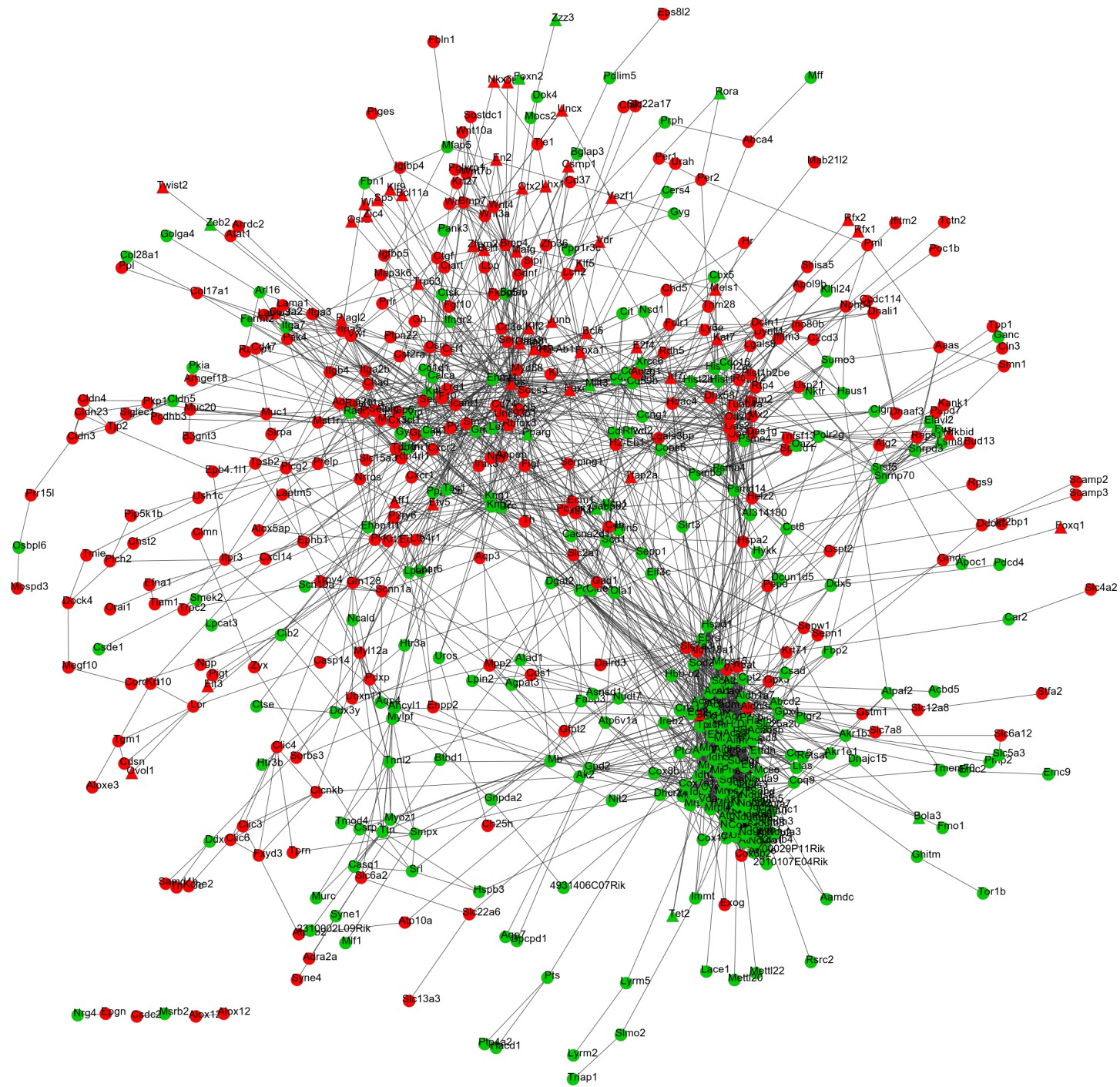

**Figure 2** **The protein–protein interaction network.** The red and green nodes represent the upregulated and downregulated genes, respectively. Triangle, transcription factors; circular, mRNA.

**Table 2  Hub genes in the protein–protein interaction network.**

| Gene_Symbol | Degree | Gene_Symbol | Betweenness | Gene_Symbol | Closeness |
|---|---|---|---|---|---|
| Uqcrc2 | 57 | Ehmt1 | 17581.75493 | Sod2 | 0.000195886 |
| Sdhb | 55 | Fos | 16050.50737 | Eprs | 0.000194932 |
| Atp5 h | 53 | Icam1 | 9986.745004 | Fos | 0.000194515 |
| Ndufs3 | 52 | Sod2 | 9941.869934 | Icam1 | 0.000193573 |
| Uqcr11 | 48 | Eprs | 9124.185173 | Pdk4 | 0.000192976 |
| Ndufs2 | 48 | Pdk4 | 7466.1032 | Pparg | 0.000192827 |
| Ehmt1 | 47 | Cav1 | 7458.835522 | Dld | 0.000192641 |
| Dld | 47 | Helz2 | 7173.879054 | Aldh18a1 | 0.000192345 |
| Cyc1 | 47 | Cops5 | 6791.408479 | Tpi1 | 0.000191939 |
| Pmpcb | 46 | Dld | 6500.567441 | Foxo1 | 0.000191608 |
| Ndufa9 | 46 | Gm128 | 6201.287809 | Ehmt1 | 0.000191168 |
| Fos | 46 | Tpi1 | 6150.610424 | Socs3 | 0.000190949 |
| Suclg1 | 45 | Pparg | 6092.823564 | Hspd1 | 0.000190767 |
| Ndufv2 | 44 | Aldh18a1 | 5723.478434 | Aldh3a1 | 0.000190404 |
| Ndufa8 | 44 | Tac1 | 5073.412043 | Aldh1a7 | 0.000190259 |
| Ndufb5 | 43 | Gldc | 4915.109074 | Sirt3 | 0.00019015 |
| Acadvl | 43 | Acadvl | 4870.244131 | Helz2 | 0.000190042 |
| Uqcr10 | 42 | Mb | 4867.912647 | Ccng1 | 0.00018997 |
| Ndufa5 | 42 | Bmp4 | 4710.157376 | Acadvl | 0.000189934 |
| Uqcrh | 41 | Foxo1 | 4510.416535 | Dbt | 0.000189934 |

**Table 3  Significant modules screened from the protein–protein interaction network.**

| Cluster | Score (Density*#Nodes) | Nodes | Edges | Node IDs |
|---|---|---|---|---|
| 1 | 15.692 | 39 | 612 | Atp5j2, Ndufv2, Ndufs8, Ndufs2, Sdhb, Sdhd, Ndufs4, Ndufs3, Ndufa8, Ndufc2, Uqcrc2, Cyc1, Etfb, Atp5 h, Atp5e, Uqcr10, Uqcr11, Pmpcb, Ndufb5, Ndufb9, Ndufa9, Ndufa5, Suclg1, Ndufb8, Ndufb4, Ndufb2, Uqcrh, Cox6c, Ndufc1, Ndufa1, 1700029P11Rik, Ndufa7, Ndufa3, Ndufb3, Atp5f1, Usmg5, Cycs, Cox6b1, Cox7b |
| 2 | 6.241 | 58 | 362 | Cpt2, Slc25a20, Ech1, Acadl, Acsl1, Aldh1a7, Aldh6a1, Aldh3a1, Acsl5, Acat2, Acadsb, Eci1, Echs1, Idh1, Dbt, Acadvl, Hadhb, Eci2, Acaa2, Acadm, Etfdh, Pcca, Pccb, Etfa, Dld, Aco2, Idh3g, Hspe1, Mrps15, Helz2, Irgm2, Rtp4, Mx2, Lgals3bp, Figf, Pcyox1l, Ola1, Ecm1, Sepp1, Serping1, Itih4, Kng2, Kng1, Mrpl20, Mrps28, Ptcd3, Mrpl30, Mrpl42, Mrpl53, Mrpl9, Mrpl27, Mrpl12, Dhx58, Oasl2, Oas2, Oasl1, Irf7, Fh1 |

kinase receptor activated by native collagen, induced deterioration of the supporting cells and consequently interfere with mechanical properties of the organ of Corti, leading to a severe decrease in auditory function. *Cai et al. (2012)* proved that exposure to an intense noise for 2 h caused site-specific changes in expression levels of genes from adhesion families in the apical (upregulated: Sell, Thbs1, Itgae, Icam1, and Itga5) and the basal (upregulated: Itga3, Itgb2, Selp, Sele, Cdh1, and Cdh2) sections of the sensory epithelium

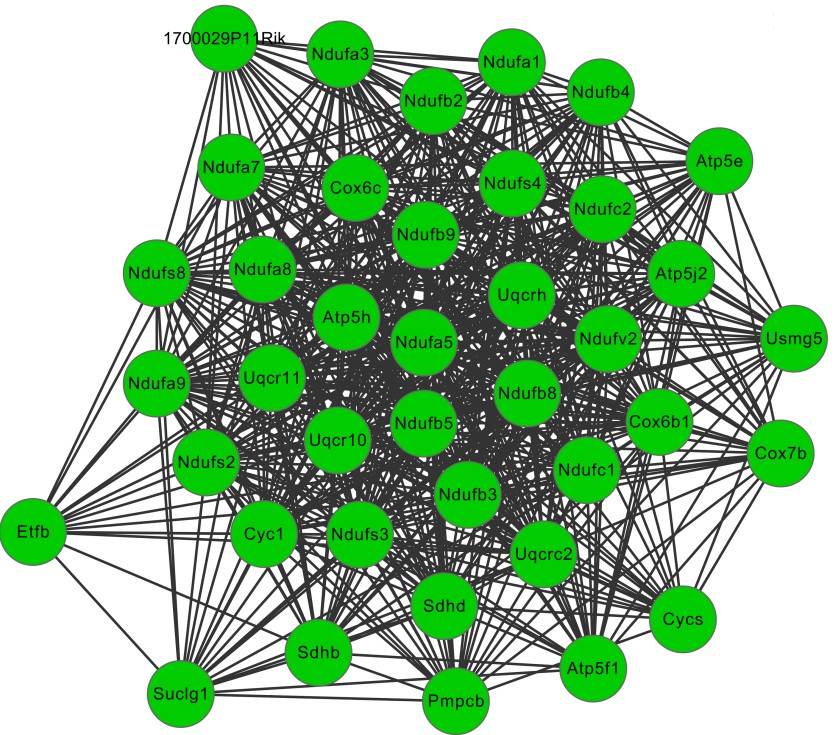

**Figure 3  The significant module 1 extracted from the protein–protein interaction network.** The green nodes represent downregulated genes.

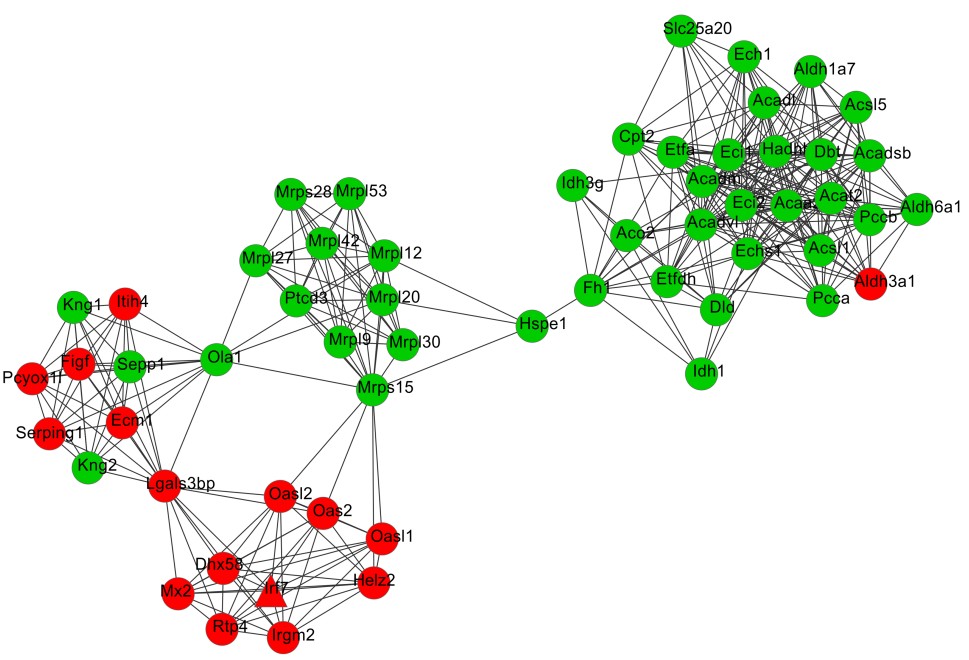

**Figure 4  The most significant module 2 extracted from the protein–protein interaction network.** The red and green nodes represent the upregulated and downregulated genes, respectively.

**Table 4   KEGG pathways for differentially expressed genes in module analysis.**

| Module | Term | *P*-value | Genes |
|---|---|---|---|
| 1 | mmu00190:Oxidative phosphorylation | 1.80E−55 | UQCRC2, NDUFB3, ATP5E, NDUFB4, NDUFB5, NDUFB8, NDUFB9, COX7B, CYC1, NDUFB2… |
| | mmu05012:Parkinson's disease | 2.20E−54 | UQCRC2, NDUFB3, ATP5E, NDUFB4, NDUFB5, NDUFB8, NDUFB9, COX7B, CYC1, NDUFB2… |
| | mmu05010:Alzheimer's disease | 9.89E−52 | UQCRC2, NDUFB3, ATP5E, NDUFB4, NDUFB5, NDUFB8, NDUFB9, COX7B, CYC1, NDUFB2… |
| | mmu05016:Huntington's disease | 4.97E−50 | UQCRC2, NDUFB3, ATP5E, NDUFB4, NDUFB5, NDUFB8, NDUFB9, COX7B, CYC1, NDUFB2… |
| | mmu04932:Non-alcoholic fatty liver disease (NAFLD) | 1.49E−45 | UQCRC2, NDUFB3, NDUFB4, NDUFB5, NDUFB8, NDUFB9, COX7B, CYC1, NDUFB2, UQCR10… |
| | mmu01100:Metabolic pathways | 1.92E−27 | UQCRC2, NDUFB3, ATP5E, NDUFB4, NDUFB5, NDUFB8, NDUFB9, COX7B, CYC1, NDUFB2… |
| | mmu04260:Cardiac muscle contraction | 3.23E−08 | UQCRC2, UQCR10, UQCR11, UQCRH, COX7B, CYC1, COX6B1, COX6C |
| | mmu00020:Citrate cycle (TCA cycle) | 0.008597 | SDHB, SUCLG1, SDHD |
| 2 | mmu00071:Fatty acid degradation | 1.63E−18 | ECI1, ECI2, ACAA2, ACADSB, CPT2, ACADM, ECHS1, ACADL, ACAT2, HADHB… |
| | mmu01212:Fatty acid metabolism | 1.89E−14 | ACADVL, ACAA2, ACADSB, ACSL1, ACADM, CPT2, ECHS1, ACADL, ACAT2, ACSL5… |
| | mmu00280:Valine, leucine and isoleucine degradation | 4.29E−14 | ACAA2, ALDH6A1, DBT, ACADSB, ACADM, DLD, ECHS1, ACAT2, PCCB, PCCA… |
| | mmu01200:Carbon metabolism | 9.82E−11 | ALDH6A1, ACADM, IDH3G, ACO2, DLD, ECHS1, IDH1, FH1, ACAT2, PCCB, PCCA |
| | mmu01130:Biosynthesis of antibiotics | 1.52E−10 | ACAA2, ACADM, ACO2, ECHS1, ACAT2, HADHB, DBT, IDH3G, DLD, IDH1, FH1, PCCB, PCCA |
| | mmu00640:Propanoate metabolism | 1.43E−07 | ALDH6A1, ACADM, ECHS1, ACAT2, PCCB, PCCA |
| | mmu01100:Metabolic pathways | 1.91E−07 | ACAA2, ALDH6A1, ACADSB, ACADM, ACO2, ECHS1, ACADL, ACAT2, ALDH3A1, HADHB… |
| | mmu00630:Glyoxylate and dicarboxylate metabolism | 9.74E−06 | ACO2, DLD, ACAT2, PCCB, PCCA |
| | mmu00020:Citrate cycle (TCA cycle) | 1.46E−05 | IDH3G, ACO2, DLD, IDH1, FH1 |
| | mmu00410:beta-Alanine metabolism | 5.01E−04 | ALDH6A1, ACADM, ECHS1, ALDH3A1 |
| | mmu03320:PPAR signaling pathway | 5.45E−04 | ACSL1, ACADM, CPT2, ACADL, ACSL5 |
| | mmu03010:Ribosome | 5.86E−04 | MRPL12, MRPS15, MRPL27, MRPL9, MRPL30, MRPL20 |
| | mmu04146:Peroxisome | 6.26E−04 | ECI2, ACSL1, ECH1, IDH1, ACSL5 |
| | mmu01210:2-Oxocarboxylic acid metabolism | 0.003631 | IDH3G, ACO2, IDH1 |
| | mmu00062:Fatty acid elongation | 0.006757 | ACAA2, ECHS1, HADHB |
| | mmu00620:Pyruvate metabolism | 0.014816 | DLD, FH1, ACAT2 |

**Notes.**
KEGG,  Kyoto Encyclopedia of Genes and Genomes.

in the cochlea. Selp and Itga5 in the basal section were positively, but Sell in the apical section was negatively correlated with greater hearing loss. In line with the above findings, our present study also identified several ECM and adherens genes to be differentially expressed, with the upregulation of LAMA1, ICAM1 and ITGA5, suggesting these genes may be underlying targets for treatment of AHL. Our hypothesis had been preliminarily demonstrated in the study of *Ramunni et al. (2006)* who found inhibition of adhesion

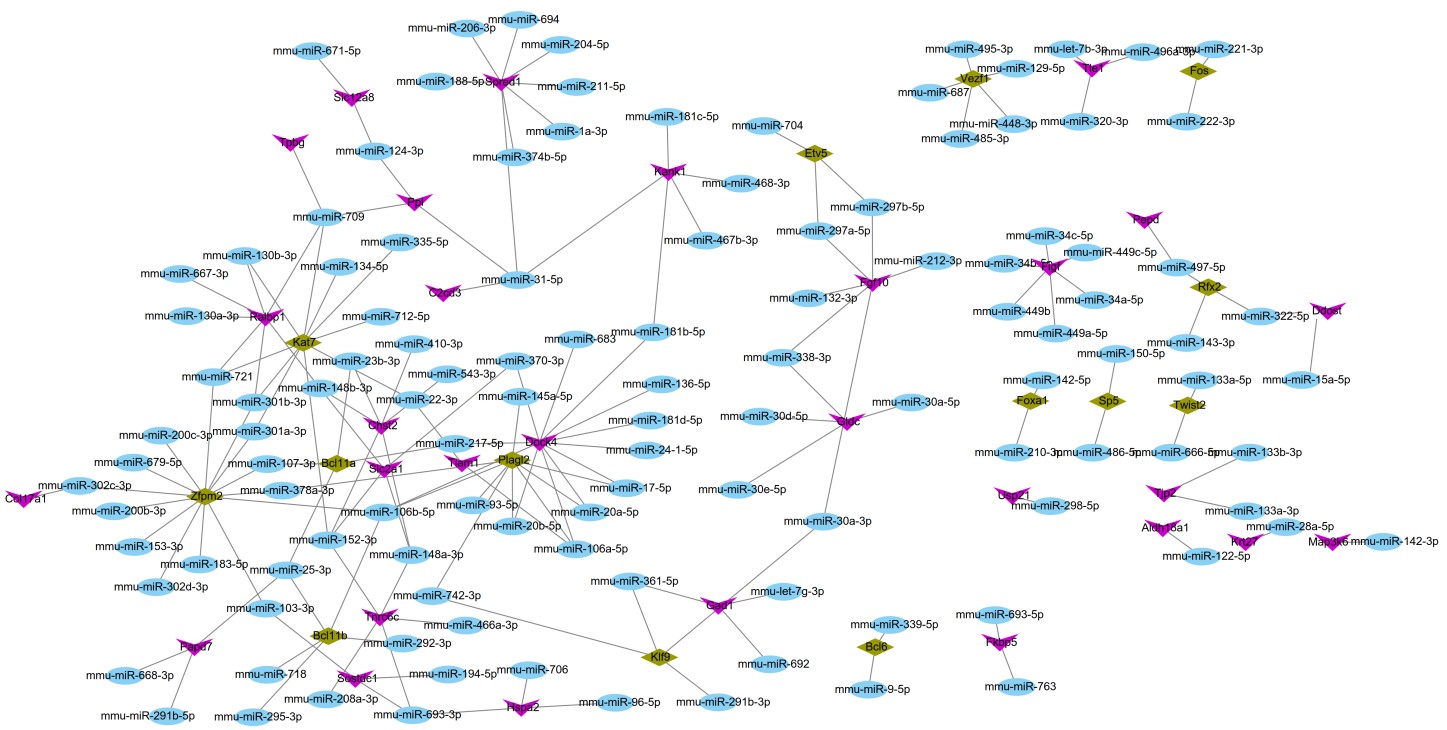

**Figure 5** **The miRNA-mRNA regulatory network for the upregulated genes.** Blue: miRNAs; purple: upregulated genes; hazel-green: upregulated transcription factors.

molecules (sE-selectin, sVCAM-1 and sICAM-1) by a single session of LDL/fibrinogen apheresis led to a complete hearing recovery.

In addition, the upregulation of adherens genes may favor the interaction between leukocytes and inner ear endothelial cells, promoting the inflammation and hearing loss (*Kanzaki et al., 2014*; *Ramunni et al., 2006*), indicating inflammation related pathways may also be a target for AHL. In accordance with our expected, TNF signaling pathway and Cytokine-cytokine receptor interaction pathways were also significantly enriched for upregulated genes (TNFSF13B; FOS; ICAM1) in this study. It had been reported that TNF-α and its receptors (TNFR1, TNFR2) were higher expressed in the cochlea of vibration- or noise-induced hearing loss (*Fuentessantamaría et al., 2017*; *Zou et al., 2005*). Use of TNF-α inhibitor preserved the hearing threshold by improvement of cochlear blood flow (*Arpornchayanon et al., 2013*). The expression of FOS was found to be dynamically changed after deafness, with lower level in the auditory cortex 15 days (compensation mechanism), but increased from 2 weeks and stabilized three months after permanent auditory deprivation in adult rats (*Pernia et al., 2017*). As a TF, FOS may participate in inflammation by regulating its target genes, such as ICAM-1, CSF1 and CCL5 which were all important inflammatory proteins for hearing loss (*Trune et al., 2015*).

PPARs are ligand-activated TFs belonging to the nuclear receptor superfamily. Extensive studies have shown that PPAR participates in various biological functions such as cell proliferation, apoptosis and differentiation by regulating its target genes (*Chung et al.,*

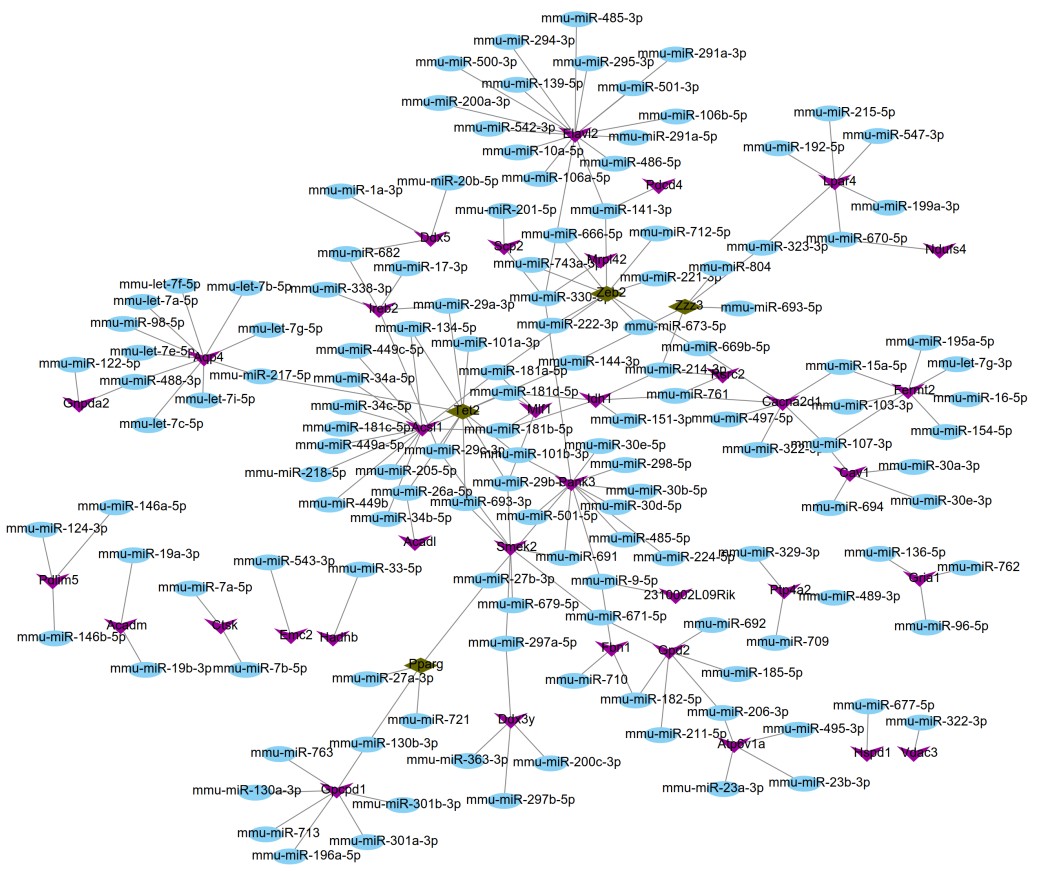

**Figure 6** **The miRNA-mRNA regulatory network for the downregulated genes.** Blue: miRNAs; purple: downregulated genes; hazel-green: downregulated transcription factors.

*2008*). For example, there is evidence to reveal that PPAR expression is inversely correlated with inflammatory cytokines IL-1β and TNF-α in aging rats (*Gelinas & Mclaurin, 2005*). The PPARγ agonist ameliorates aging-related renal and cerebral artery injuries by inhibiting the inflammatory genes, reducing ECM production, and attenuating oxidative stress (*Sung et al., 2006*; *Wang et al., 2014*; *Yang et al., 2009*). In this study, we also found PPAR $\gamma$ was downregulated in the cochlear tissues of Cmah-null mice and our PPI network showed PPARG could interact with SOD2, FOS and ICAM1, implying PPARG mediated pathways may be also a considerably important mechanism for AHL and activation of PPARG may be an underlying therapeutic method for patients with AHL, which has not been reported previously.

MicroRNAs (miRNAs) are a class of small RNAs (18–25-nucleotide) that down-regulate the expression of target genes via binding to the 3′-untranslated region (UTR) and then participate in the cellular processes. There has been evidence to indicate miRNAs participate in the pathogenesis of AHL, including miR-34a (*Huang et al., 2017*; *Pang et al., 2017*) and miR-29b (*Xue et al., 2016*). These two miRNAs were involved in AHL by regulating ROS homeostasis-related gene SIRT1 and then influencing cochlear hair cell apoptosis. However,

**Table 5  Small molecule drugs predicted by the Cmap database.**

| Cmap name | Mean | N | Enrichment | P |
|---|---|---|---|---|
| adiphenine | −0.727 | 5 | −0.958 | 0 |
| DL-PPMP | −0.672 | 1 | −0.95 | — |
| decitabine | −0.617 | 1 | −0.915 | — |
| topiramate | −0.596 | 1 | −0.899 | — |
| 5186324 | −0.578 | 1 | −0.886 | — |
| 5213008 | −0.573 | 1 | −0.883 | — |
| sulindac sulfide | −0.56 | 1 | −0.869 | — |
| BW-B70C | −0.555 | 1 | −0.865 | — |
| 5186223 | −0.553 | 1 | −0.863 | — |
| isoxicam | −0.665 | 5 | −0.852 | 0.00016 |
| cefamandole | −0.574 | 4 | −0.845 | 0.00105 |
| isoflupredone | −0.623 | 3 | −0.815 | 0.01256 |
| 5140203 | −0.5 | 1 | −0.815 | — |
| tyrphostin AG-1478 | −0.497 | 1 | −0.812 | — |
| 12,13-EODE | −0.497 | 1 | −0.812 | — |
| STOCK1N-35696 | −0.582 | 2 | −0.808 | 0.07269 |
| lisuride | −0.57 | 5 | −0.808 | 0.00058 |
| 5149715 | −0.483 | 1 | −0.801 | — |
| Prestwick-692 | −0.629 | 4 | −0.799 | 0.0032 |
| carbimazole | −0.568 | 3 | −0.792 | 0.01831 |
| PF-00539745-00 | −0.517 | 3 | −0.79 | 0.01893 |
| Prestwick-691 | −0.562 | 3 | −0.78 | 0.02195 |
| indoprofen | −0.519 | 4 | −0.779 | 0.00493 |
| 5162773 | −0.457 | 1 | −0.778 | — |
| iloprost | −0.494 | 3 | −0.759 | 0.02876 |
| 5151277 | −0.427 | 1 | −0.755 | — |
| splitomicin | −0.423 | 1 | −0.752 | — |
| Prestwick-1082 | −0.625 | 3 | −0.743 | 0.03481 |
| clorsulon | −0.567 | 4 | −0.739 | 0.00915 |
| vigabatrin | −0.539 | 3 | −0.738 | 0.03684 |
| 3-acetamidocoumarin | −0.538 | 4 | −0.736 | 0.00959 |
| thiamphenicol | −0.541 | 5 | −0.734 | 0.00274 |
| levobunolol | −0.48 | 4 | −0.731 | 0.01056 |
| oxolamine | −0.551 | 4 | −0.72 | 0.01241 |
| cinchonine | −0.494 | 4 | −0.716 | 0.01327 |
| trimethobenzamide | −0.536 | 5 | −0.714 | 0.00415 |
| atracurium besilate | −0.559 | 3 | −0.712 | 0.04853 |
| levomepromazine | −0.517 | 4 | −0.71 | 0.01456 |
| chloropyrazine | −0.492 | 4 | −0.708 | 0.0151 |
| tranexamic acid | −0.548 | 5 | −0.703 | 0.00505 |
| isometheptene | −0.517 | 4 | −0.701 | 0.01657 |

**Table 5** (*continued*)

| Cmap name | Mean | N | Enrichment | P |
|-----------|------|---|-----------|---|
| benzbromarone | −0.492 | 3 | −0.698 | 0.05574 |
| heptaminol | −0.544 | 5 | −0.687 | 0.00685 |
| trihexyphenidyl | −0.406 | 3 | −0.68 | 0.06572 |
| Prestwick-642 | −0.4 | 4 | −0.68 | 0.02304 |
| viomycin | −0.53 | 4 | −0.678 | 0.02383 |
| naringenin | −0.463 | 4 | −0.663 | 0.0291 |
| colistin | −0.415 | 4 | −0.655 | 0.03288 |
| guanabenz | −0.488 | 5 | −0.649 | 0.0131 |
| canadine | −0.442 | 4 | −0.647 | 0.0369 |
| sulmazole | −0.442 | 3 | −0.646 | 0.08926 |
| Gly-His-Lys | −0.512 | 3 | −0.642 | 0.09278 |
| terazosin | −0.444 | 4 | −0.637 | 0.04176 |
| sulfadimethoxine | −0.501 | 5 | −0.628 | 0.01852 |
| iopamidol | −0.411 | 4 | −0.622 | 0.05101 |
| PHA-00745360 | −0.432 | 8 | −0.62 | 0.00172 |
| iodixanol | −0.423 | 3 | −0.619 | 0.11656 |
| ribavirin | −0.469 | 4 | −0.614 | 0.05616 |
| atractyloside | −0.485 | 5 | −0.593 | 0.03218 |
| mycophenolic acid | −0.42 | 3 | −0.591 | 0.15331 |
| Prestwick-1103 | −0.472 | 4 | −0.59 | 0.07607 |
| aciclovir | −0.415 | 6 | −0.587 | 0.01712 |
| triflupromazine | −0.446 | 4 | −0.578 | 0.08749 |
| rifampicin | −0.415 | 4 | −0.577 | 0.08825 |
| acemetacin | −0.477 | 4 | −0.571 | 0.09443 |
| bumetanide | −0.459 | 4 | −0.57 | 0.09533 |
| josamycin | −0.407 | 5 | −0.551 | 0.05727 |
| trapidil | −0.404 | 3 | −0.544 | 0.23612 |
| proguanil | −0.463 | 3 | −0.524 | 0.27656 |

as a crucial gene identified to be associated with ROS in AHL of our study, there was no study to investigate the miRNAs that regulate PPARG in AHL. Thus, we also predicted the potential miRNAs that regulate PPARG by using the miRwalk database. As a result, miR-130b-3p, miR-27a-3p, miR-27b-3p and miR-721 were screened. miR-27b-3p has been shown to target PPARG to inhibit cell proliferation, but increase the inflammatory response to promote cell apoptosis (*Lee et al., 2012*). Nevertheless, there were no studies on the relationship between PPARG and others miRNAs in cell apoptosis, which may be our future research direction.

Furthermore, we also identified the potential drugs for inhibiting PPARG, consisting of the most negatively correlated adiphenine, DL-PPMP, decitabine and topiramate. Several studies have demonstrated topiramate could attenuate oxidative damage, inflammation and neuronal cell death (*Motaghinejad & Shabab, 2016*; *Tian et al., 2015*), indicating topiramate may also be an underlying drug for PPARG-related AHL.

However, there were some limitations in this study. First, the sample size in the microarray data GSE70659 was small. Another microarray or sequencing experiments should be performed to further screen crucial mechanisms for AHL. Second, we only preliminarily identified the AHL-related genes, miRNAs and drugs. Additional *in vivo* and *in vitro* experiments (PCR, Western blotting, knockout and overexpression design) are necessary to confirm their expression and their functions.

## CONCLUSION

Our present study preliminarily reveals Cmah deficiency may lead to AHL by downregulating PPARG, which may then induce the higher expressions of ECM and adhesion (ICAM1) and pro-inflammatory (FOS, TNFSF13B), but lower expression of anti-oxidative genes (SOD2). Downregulation of miR-130b-3p, miR-27, miR-721 and the use of topiramate may be potential treatment strategies for ALH by upregulating PPARG.

### Funding
This work was supported by National Natural Science Foundation of China (Grant No. 81570924). The funders had no role in study design, data collection and analysis, decision to publish, or preparation of the manuscript.

### Grant Disclosures
The following grant information was disclosed by the authors:
National Natural Science Foundation of China: Grant No. 81570924.

### Competing Interests
The authors declare there are no competing interests.

### Author Contributions
- Juhong Zhang conceived and designed the experiments, performed the experiments, analyzed the data, contributed reagents/materials/analysis tools, prepared figures and/or tables, authored or reviewed drafts of the paper, approved the final draft.
- Na Wang performed the experiments, prepared figures and/or tables, approved the final draft.
- Anting Xu conceived and designed the experiments, authored or reviewed drafts of the paper, approved the final draft.

### Data Availability
The raw data is available in Supplemental Information 1.

### Supplemental Information
Supplemental information for this article can be found online at http://dx.doi.org/10.7717/peerj.6856#supplemental-information.

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
