# Peer review of "Cmah deficiency may lead to age-related hearing loss by influencing miRNA-PPAR mediated signaling pathway"

_PeerJ, doi:10.7717/peerj.6856_

## Round 0.1 · original submission · Major Revisions

Please address all the concerns of all the reviewers and revise your manuscript accordingly.

·

Basic reporting

The manuscript is clear with the usage of professional English.

sufficient literature has been cited.

The article has professional structure, figures, and tables.

the manuscript is heavy on hypothetical explanations.

Experimental design

Primary research is based on older data.

The research question is not very well defined, seems like the author wants to cover too many topics in one manuscript.

Rigorous investigations are not performed, most of the data os hypothetical with not experimental investigations.

The methods are described in sufficient detail.

Validity of the findings

The investigation brings a lot of hypothesis with no experimental data.

The data needs more work and investigation.

The conclusions are very speculative.

Additional comments

1. Figures 1. Shows 3 datasets for Cmah. Null and WT experiments. Are these 3 datasets are biological replicates? Please mention this in the figure legend. What is the main difference between the 3 datasets?
2. Line 65, the data cited is old, it dates back to 2015, is there an updated figure? If so, please cite that data.
3. What is TF? Is its transcription factor, if so please mention the details in the manuscript?
4. The manuscript is very speculative with a lot of hand waving. It would be advisable if some solid experimental data can be provided to confirm at least some of the proposed hypotheses.

Reviewer 2 ·

Basic reporting

The English language used in this manuscript need to be improved and polished. This reviewer found there are many long sentences with convolved grammar structure, which makes it challenging to read. For example:
1. In line 101-105, this sentence is very long and contains at least two messages: a) the authors aimed to screen more crucial gene and pathways using the published microarray data; b) the authors constructed network maps in order to find potential treatments for AHL. It will be much clearer if the authors can separate this long sentence into two and convey a concise message in each one.
2. In line 285-288, the authors adopted another super long sentence "It has been reported that..., but..., but..., which...". It is very unclear and reads uncomfortable when using two "but" to connect a sentence. Please conform to professional standards of English expression.

Moreover, in the Abstract session, the authors utilized several abbreviations without spelling out at the first use, such as KEGG, ECM, PPAR. Please spell out the full names without bothering readers to look it up.

In the second paragraph of discussion, the authors spent a whole page to describe ECM is essential for structural and functional integrity of organs. This reviewer found it's too distracting and unnecessary. The discussion part should be focus on the reflection on the findings and how to interpret the result.

Another small suggestion: according to PeerJ standard format, headings in structured abstracts should be bold and followed by a period. Each heading should begin a new paragraph. For example:
"Background. The background section text goes here. Next line for new section."
Please review PeerJ template for standard formatting (https://peerj.com/about/author-instructions/#standard-sections).

One small typo, in line 224, it should be AHL but not ALH.

Experimental design

The research questions is well defined: to explore the mechanism of Age-related Hearing Loss (AHL) by using microarray analysis in the Cmah deficiency animal model.

The methods were described with sufficient detail and reference.

Validity of the findings

The microarray data were collected from the Gene Expression Omnibus database, not from the authors' own experiment.

The data analysis and conclusions are well stated and linked to the original research question.

Additional comments

Generally speaking, the data analysis and conclusions are clear. However, the english language can be much more concise and crisp to convey straightforward ideas.

Reviewer 3 ·

Basic reporting

1. This manuscript was written in professional and technically correct English with some minor phrasing problems in the Introduction. To ensure a better understanding of your article by a broader audience the English language in the introduction can be improved and some examples include line 81-82, 89-92.

2. This article investigated CMP-Neu5Ac hydroxylase (Cmah) deficiency induced aging-related hearing loss, hence in the introduction (line 93-94), the authors could provide more background information on Cmah and its disruption in biological processes.

3. I thank the authors for providing all the required raw data and relevant figures in this manuscript. To ensure a better understanding of the study, authors could consider improving the figure legend of Figure 1 by adding an explanation of features in the figure and a brief statement of the results that can be gleaned from this figure.

Experimental design

In this work, the authors aimed to advance the understanding of the mechanism of age-related hearing loss and benefit the therapeutic interventions using the Camh deficiency mouse model. Based on the work published by Deug-Nam et al. the investigators identified additional differentially expressed genes and pathways by constructing protein-protein interaction (PPI) network and miRNA regulatory network. The finding of this work has its value to the field and may potentially fill the gap in understanding of AHL.

Experiments were well designed with proper control. Differentially expressed genes (DEG) were subjected to an intensive investigation to identify their biological functions.

Validity of the findings

The Data presented in this work is statistically sound and properly controlled. The authors should consider the following comments for revision before publication.

1. This study is based on the work published by Deug‐Nam Kwon in 2015. Deug‐Nam Kwon identified 631 up-regulated genes, 729 down-regulated genes in Cmah-null mice derived cochlear tissues compared to control mice. In this work, a total of 700 DEGs were identified with 449 up-regulated and 350 down-regulated from the same set of microarray data. Please specify the reasons for this difference.

2. In figure 6, it has been predicted that PPARG can be regulated by mmu-miR-221-3p, mmu-miR-130b-3p, mmu-miR-27a-3p, mmu-miR-27b-3p and mmu-miR-721. There is no further evidence in this article showing that miRNA-27a-3p is the only or major miRNA regulating PPARG in AHL disease mouse model, I suggest that you improve the description at lines 285- 288 to provide more justification for your conclusion or consider modifying the paper title and the conclusion in line 303-304 accordingly.

3. Line 220-221, the authors concluded small molecules like adiphenine, DL-PPMP, decitabine, and topiramate could potentially reverse the expression of PPARG in AHL. To help broader readers to understand this study, please provide more detail on the data collected from the Connectivity Map. Examples include 1. The genome-wide transcriptional expression data in Connectivity Map was collected from wild-type human cell lines or mutated cell lines? 2. Table 5 listed 69 small molecules found from the CAMP search. Since all DEGs in PPI network were uploaded into CMAP it would be helpful to group these small molecules according to their features such as targeted genes or inhibitors/activators.

---

## Round 0.2 · accepted · Accept

All critical issues were adequately addressed and the manuscript was appropriately revised.

# ·

Basic reporting

No comments.

Experimental design

No comments.

Validity of the findings

No Comments.

Additional comments

Considering the changes made in the present version of the manuscript, as suggested by the reviewers, I can now recommend this manuscript for publication.

Reviewer 2 ·

Basic reporting

The manuscript after revision has addressed all my concerns. This is ready to publish.

Experimental design

The manuscript after revision has addressed all my concerns. This is ready to publish.

Validity of the findings

The manuscript after revision has addressed all my concerns. This is ready to publish.

Additional comments

The manuscript after revision has addressed all my concerns. This is ready to publish.